# Phytochrome B and phytochrome-interacting-factor4 modulate tree seasonal growth in cold environments

Bo Zhang [1] ✉, Keh Chien Lee[1], Laura García Romañach [1], Jihua Ding [1,2], Alice Marcon[1] & Ove Nilsson [1] ✉

Plants that live at high latitudes and altitudes must adapt to growth in cold environments. Trees survive freezing winter conditions by ceasing growth and forming protective winter buds at the end of the growing season. To optimize growth and adaptation, the timing of growth cessation and bud set is critical. Like the well-studied *Populus* species (poplars, aspens, cottonwoods), many trees respond to the shortening photoperiods of fall to induce growth cessation. Temperature also has a role in this process, but the mechanism is unknown. Here, we show that the *PHYTOCHROME B* (*PHYB*)-*PHYTOCHROME INTERACTING FACTOR4* (*PIF4*) module controls the interplay between photoperiod cues and temperature to prevent premature growth cessation and bud set at cooler temperatures. *PHYB* is essential for the ability of aspen trees to maintain growth under lower temperatures in permissive long days. This is mediated through PIF4, which promotes growth cessation, specifically in response to low temperatures rather than to changes in photoperiod. PIF4 can directly bind to the promoter region of the vegetative growth marker gene *FLOWERING LOCUS T2* (*FT2*). In contrast to annual plants, it does so to suppress its transcription. Furthermore, lower temperatures can suppress PIF4 function at the transcriptional and protein levels to prevent premature growth cessation. These data show how poplar trees balance the antagonistic roles of *PHYB* and *PIF4* to optimise the timing of growth cessation and bud set in cold environments, and this has been achieved with contrasting mechanisms compared to the annual plant model.

Plants experience temperature variations at both daily and seasonal levels[1,2]. Recently, because of the concern of global warming, plant thermomorphogenesis has been increasingly studied, where plants respond to mild ambient temperature elevations within a range of 22–30 °C[3,4]. On the other hand, plants from high latitude and altitude regions must adapt to relatively low temperatures during the growing season. For instance, the average temperatures during the Swedish summer (June-August) vary between 12 and 16 °C (https://www.smhi.se/). During this period, trees first experience active growth, then growth cessation. Since the temperature can potentially affect growth cessation and bud set, and temperature is a much less reliable proxy for the time of year than day length, the tree needs a mechanism to prevent premature growth cessation and bud set in response to lower summer temperatures under permissible Long-Day (LD) conditions[5]. So far, how trees adapt their growth to milder alterations in temperature remains largely unknown. *Populus* species

[1]Umeå Plant Science Centre, Department of Forest Genetics and Plant Physiology, Swedish University of Agricultural Sciences, Umeå, Sweden. [2]Present address: College of Horticulture and Forestry, Huazhong Agricultural University, Wuhan, China. ✉e-mail: bo.zhang@slu.se; ove.nilsson@slu.se

have a wide geographical distribution, growing up to 3000 metres altitude[6] and northwards of the Arctic Circle[7], making them ideal for studying the genetic mechanism of climate adaptation in trees.

Temperature has been reported to control seasonal growth in some woody species, including rowans, dogwoods, apples, and pears[8–10]. In other species, such as poplars, willows, and birches, photoperiod is believed to play a dominant role in controlling growth cessation in autumn[7,11,12]. However, recent studies have shown that temperature can also significantly influence growth cessation and bud set in poplar trees[5,10]. While warm temperatures could accelerate growth cessation in different poplar hybrids[13], low temperatures were shown to delay SD-induced growth cessation in hybrid aspen[14], which has also been observed in field trials of various aspen genotypes[15]. The molecular mechanisms of photoperiod-controlled seasonal growth in trees have been well investigated[11]. In *Populus* trees, the activity of *FLOWERING LOCUS 2* (*FT2*) genes is crucial for maintaining vegetative growth[16,17]. Recent studies have shown that various transcriptional factors suppress *FT2* gene expression during SD treatment, resulting in growth cessation[18–22]. So far, the molecular mechanism for the influence of temperature on seasonal growth is still missing.

Recently, some significant breakthroughs have been made in our understanding of plant thermomorphogenesis signalling[1,3], which opens new avenues to study the regulation of low-temperature growth. Here, we show a central role for the PHYB-PIF4 module in maintaining the growth of poplar trees under permissive daylengths in cold environments commonly found at high latitude or altitude regions[23]. We show that PHYB is required for the growth of poplar trees at lower temperatures. *PHYB* regulates *PIF4*, which has a specific role in responding to low temperatures in a post-translational manner. We also found that PIF4 suppresses the transcription of the central growth regulator *FT2* in poplar trees, contrary to its counterpart in annual plants. Our findings suggest that the PHYB-PIF4 module plays a significant role in temperature sensing during the seasonal growth in poplar trees.

## Results

### PHYB is required for vegetative growth at lower temperatures

In Arabidopsis, Phytochrome B (phyB) can function as a thermosensor to affect hypocotyl growth[24,25]. The finding that the active phytochrome Pfr state is maintained at higher levels at low temperatures suggests that phyB might be important in controlling plant low-temperature growth[24]. To study if phyB plays a role in the temperature regulation of aspen tree seasonal growth, and because *PHYB1 PHYB2* double knockout aspen trees do not survive[21], we generated *PHYB* RNAi transgenic aspen trees, in which both *PHYB* paralogs, *PHYB1* and *PHYB2*, were downregulated (Supplementary Fig. 1a). We characterised the growth of *PHYBRNAi* trees at 15 °C compared to 21 °C under 18 h light/6 h dark LD conditions. At the early stages of growth, similar to Arabidopsis, *PHYBRNAi* lines displayed a constitutively enhanced height-growth at 21 °C and 15 °C compared to the WT trees (Fig. 1a). Surprisingly, after 50 days of 15 °C LD, the two independent *PHYBRNAi* transgenic lines stopped growing while they continued to grow at 21 °C (Fig. 1a, b, Supplementary Fig. 1b). In contrast, the WT trees never stopped growth at both temperatures (Fig. 1a, b, Supplementary Fig. 1b). Furthermore, the *PHYBRNAi* plants formed hard apical buds resembling short-day (SD) induced buds (Fig. 1b, Supplementary Fig. S1c)[26]. *FLOWERING LOCUS T2* (*FT2*) is required for vegetative growth in poplar trees[16,17,27]. Consistent with the growth cessation phenotypes, we found that the diurnal expression levels of the two *FT2* paralogs, *FT2a* and *FT2b*, were significantly downregulated in the *PHYB* knockdown plants (Fig. 1c, d). These findings suggest that *PHYBs* are essential for low-temperature vegetative growth by promoting *FT2* transcription.

To further validate its role in *FT2* transcription, we generated *PHYB* overexpressing (OE) transgenic plants (*35S::PHYB2-GFP*). Since

*FT2* transcriptional levels are suppressed in SD, resulting in growth cessation and bud formation[16], we subjected trees to SD to investigate whether over-expression of *PHYB2* could alter the SD-induced growth cessation process. The *PHYB2 OE* trees displayed a delayed growth cessation at 21 °C compared to WT, coupled with a significantly enhanced expression of *FT2* (Fig. 1e, Supplementary Fig. 1d, h, i). However, this difference was abolished at 15 °C for both independent *PHYB2 OE* transgenic lines (Fig. 1f, Supplementary Fig. 1e). A similar change was also observed in the *PHYBRNAi-12* plants, which ceased growth seven days earlier than WT at 15 °C but 14 days earlier at 21 °C (Supplementary Fig. 1f, g). These findings are consistent with the potentially higher levels of active PfrB at low temperatures, which could compromise the delayed growth cessation caused by its transcriptional alterations. We also found that at low temperatures, aspen trees tend to increase *PHYB* transcriptional levels at dawn (Supplementary Fig. 1a). Taken together, the unexpected growth cessation observed on *PHYB* RNAi plants at 15 °C LD suggests that the increase in PfrB that occurs at lower temperatures might be critical to prevent premature growth conditions when temperature has decreased at still permissive daylengths.

### PIF4 suppresses low-temperature vegetative growth in LD

In Arabidopsis, the PHYTOCHROME INTERACTING FACTOR 4 (PIF4) transcription factor acts as a hub in ambient-temperature signalling[3,28,29]. We wanted to know whether *PIF4* is involved in the *PHYB*-controlled low-temperature response in aspen trees. Like the plants growing at 21 °C LD[21], the *PIF4RNAi* lines height was significantly reduced compared to the WT at 15 °C (Fig. 2a, Supplementary Fig. 2a), and *PIF4RNAi* suppressed the *PHYBRNAi-12* height growth in the *PHYB/PIF4* double RNAi lines (Fig. 2a, Supplementary Fig. 2a). The *PHYB/PIF4* double RNAi plants ceased growth significantly later than the *PHYBRNAi* plants at 15 °C (Fig. 2b, Supplementary Fig. 2b). The diurnal expression analysis showed similar overall levels of the two *FT2* paralogs in the *PIF4RNAi* plants compared to WT (Fig. 2c, Supplementary Fig. 2c). Interestingly, the *PHYB/PIF4* double RNAi plants showed significantly enhanced *FT2b* expression levels in the early morning (ZT6), late afternoon (ZT14), and early evening (ZT18,20) compared to the *PHYBRNAi* plants (Fig. 2c). These results show that in terms of *FT2* regulation, *PIF4RNAi* partially suppresses the *PHYBRNAi* phenotype and suggest that, as a downstream target of *PHYB*, *PIF4* is a negative regulator of *FT2* transcription during *PHYB*-controlled low-temperature vegetative growth in LD.

### PIF4 specifically mediates SD-induced growth cessation at low temperature

Considering the effects of *PIF4RNAi* and *PHYBRNAi* in low-temperature LD, we then investigated whether a similar genetic interaction exists during low-temperature in SD. As previously reported[21], *PIF4RNAi* and WT plants ceased growth (bud score 2[26]) at 21 °C at the same time, while *PHYBRNAi* and *PHYB/PIF4* double RNAi plants showed the same earlier growth cessation compared to WT (Fig. 2d), suggesting that PIF4 has no role during these conditions. In contrast, at 15 °C SD, *PIF4RNAi* plants showed a significantly delayed growth cessation compared to WT (Fig. 2e), while the *PHYB/PIF4* double RNAi plants showed a growth cessation phenotype intermediate between *PHYBRNAi* and WT (Fig. 2e). Consistent with the phenotypic data, the diurnal expression analysis showed significantly higher levels of the two *FT2* paralogs at dusk in the *PIF4RNAi* plants than in the WT (Fig. 2f, Supplementary Fig. 2d) while the *PHYB/PIF4* double RNAi showed a significantly enhanced expression of the predominantly expressed *FT2b* paralog at ZT18 compared to the *PHYBRNAi* plants (Fig. 2f), but similar to the WT expression level. This suggests that PIF4 has a specific role in promoting SD-induced growth cessation in aspen trees at low temperatures.

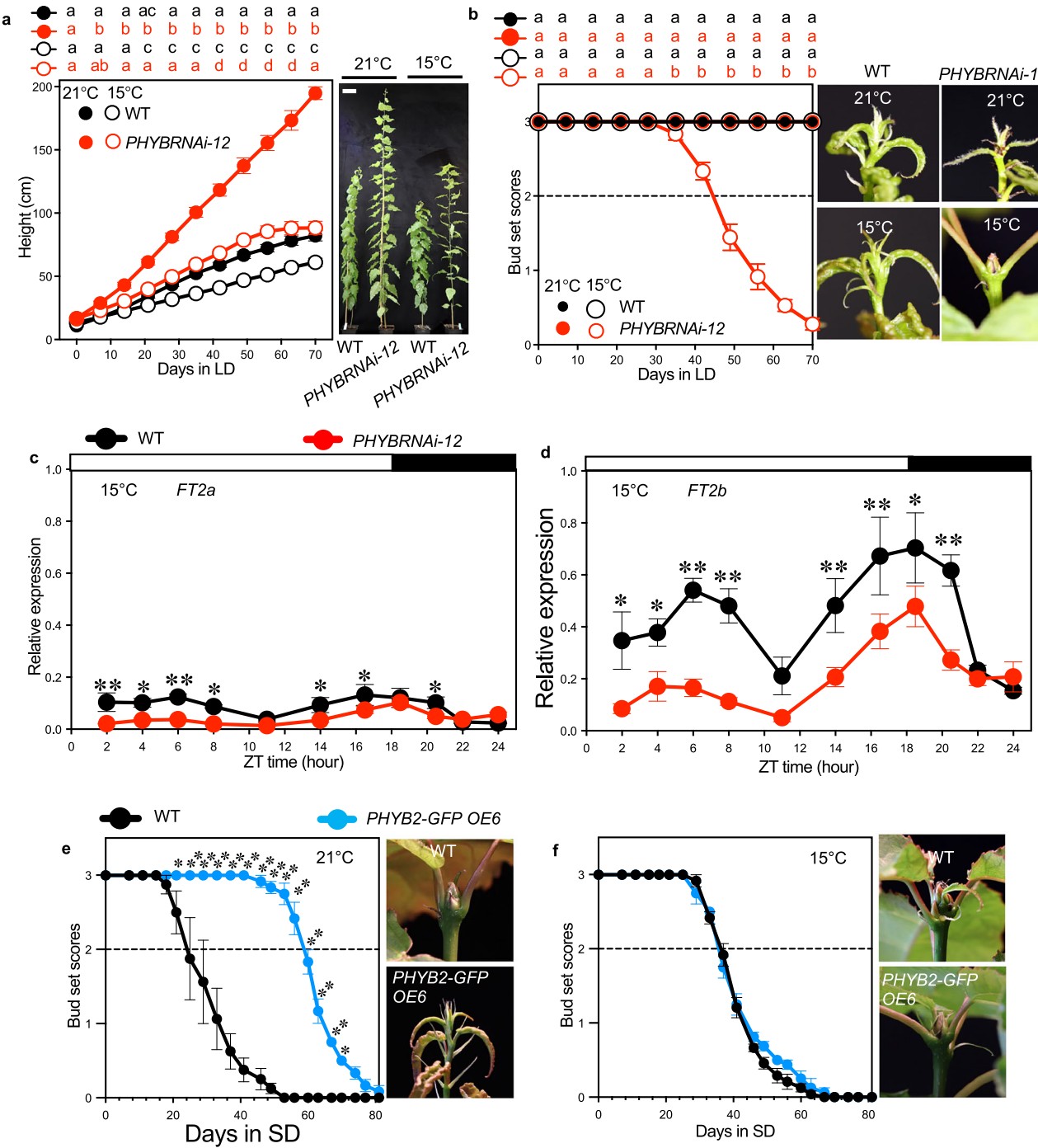

**Fig. 1 | At low temperatures, *PHYBs* are essential to maintain growth. a** Height growth of WT and *PHYBRNAi-12* plants in LD at 21 °C and 15 °C. Representative pictures of whole trees are shown in the right panel. Bars indicate a length of 10 cm. Pictures were taken after 70 days in LD at 21 °C and 15 °C. **b** Bud set scores of WT and *PHYBRNAi-12* plants in LD at 21 °C and 15 °C. The right pictures show shoot apices of WT and *PHYBRNAi* plants after 70 days in LD at 21 °C and 15 °C. (**c, d**) Diurnal relative gene expression of *FT2a* and *FT2b* in leaves of WT and the *PHYBRNAi-12* trees grown for eight weeks in LD at 15 °C. **e, f** Bud set scores of WT and the *PHYB2-GFP OE6* plants in SD at 21 °C and 15 °C. Representative pictures of the shoot apices are shown in the right panel. Pictures were taken when bud set scores of WT plants reached stage 0.5 (green closed apical bud)[26]. The dotted line in b,e,f marked bud set score 2, which indicates the stage of growth cessation[26]. Data are presented as mean ± SEM (*n* = 9 for **a**, **b**; *n* = 3 for **c**, **d**, **e**, **f**). Following ANOVA, pairwise comparisons were performed using Fisher's least significant difference (LSD) test without multiple testing correction at each time point. The statistically significant differences are indicated by non-matching letters in **a,b** ($p < 0.05$). Asterisks in **c**, **d**, **e** indicate the levels of statistical significance, *$p < 0.05$, **$p < 0.01$. No statistical differences were detected in (**f**). See also Supplementary Fig. 1 and Supplementary Table 1. Source data are provided as a Source Data file.

## PIF4 is a suppressor of *FT2* transcription in aspen trees

To further validate PIF4's role in suppressing vegetative growth and promoting SD-induced growth cessation at low temperatures, we investigated *PIF4a* overexpressing plants, in which C-terminal YFP-tagged *PIF4a* was ectopically expressed from the *35S* promoter (Supplementary Fig. 3a). All *PIF4a* overexpressing plants showed severely arrested growth in soil (Supplementary Fig. 3b). After five weeks of soil growth, WT plants reached a height of 100 cm, while all

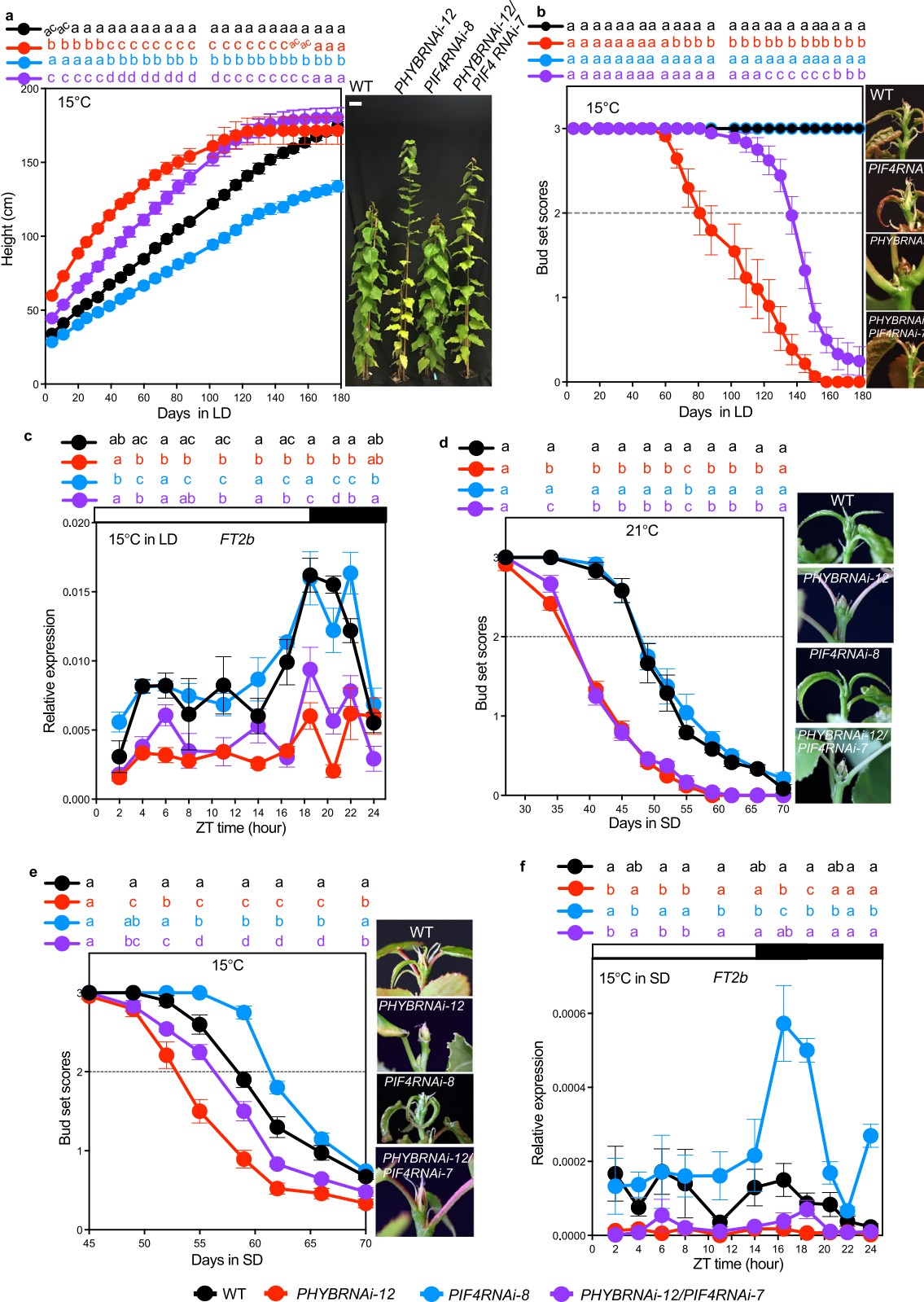

the *PIF4a* overexpressing plants were about 10–20 cm and stopped growth prematurely because of poor rooting ability (Supplementary Fig. 3c). To obtain soil-grown *PIF4a* overexpressing plants, we grafted the *PIF4a-YFP OE3* scions onto WT rootstocks (Fig. 3a). The *PIF4a-YFP OE3* grafted plants ceased growth after only ten days at 15 °C LD. After 20 days, they stopped growth completely and formed apical buds, while the WT self-grafted plants continued to grow (Fig. 3b).

Strikingly, we also found that in 15 °C LD, the *PIF4a* overexpressing plants, similar to *PHYB1 PHYB2* double KO plants[21], went into growth cessation and formed buds even in tissue culture (Supplementary Fig. 3d). The *FT2a* and *FT2b* expression was strongly downregulated in the *PIF4a-YFP OE3* plants (Fig. 3c), suggesting that *PIF4* in aspen trees is a suppressor of *FT* transcription, opposite to what has been seen in Arabidopsis[30].

**Fig. 2 | *PIF4* specifically regulates growth at low temperatures. a** Height growth of WT, *PHYBRNAi-12*, *PIF4RNAi-8*, and *PHYBRNAi-12/PIF4RNAi-7* plants in 15 °C LD. Representative pictures of whole trees are shown in the right panel. Bars indicate a length of 10 cm. Pictures were taken after 80 days in LD at 15 °C. **b** Bud set scores of WT and RNAi plants in 15 °C LD. The right pictures show the shoot apices after 140 days in 15 °C LD. **c** Diurnal relative gene expression of *FT2b* in 15 °C LD. Leaf samples were collected after 56 days in LD. **d, e** Bud set scores of WT and RNAi plants during the SD-induced growth cessation at 21 °C and 15 °C. Representative pictures of the shoot apices are shown in the right panel. Pictures were taken when the bud set scores of plants started reaching stage 0.5 (green closed apical bud[26]). **f** Diurnal relative gene expression of *FT2b* in 15 °C SD. Leaf samples were collected after 7 days in SD. Data are presented as mean ± SEM (*n* = 12 for **a**, **b**; *n* = 6 for **d**, **e**; *n* = 3 for **c**, **f**). Following ANOVA, pairwise comparisons were performed using Fisher's LSD test (two-sided) without multiple testing correction at each time point. The statistically significant differences are indicated by non-matching letters (*p* < 0.05). The dotted line in (**b**, **d**, and **e**) marks bud set score 2, indicating the growth cessation stage. See also Supplementary Fig. 2 and Supplementary Table 1. Source data are provided as a Source Data file.

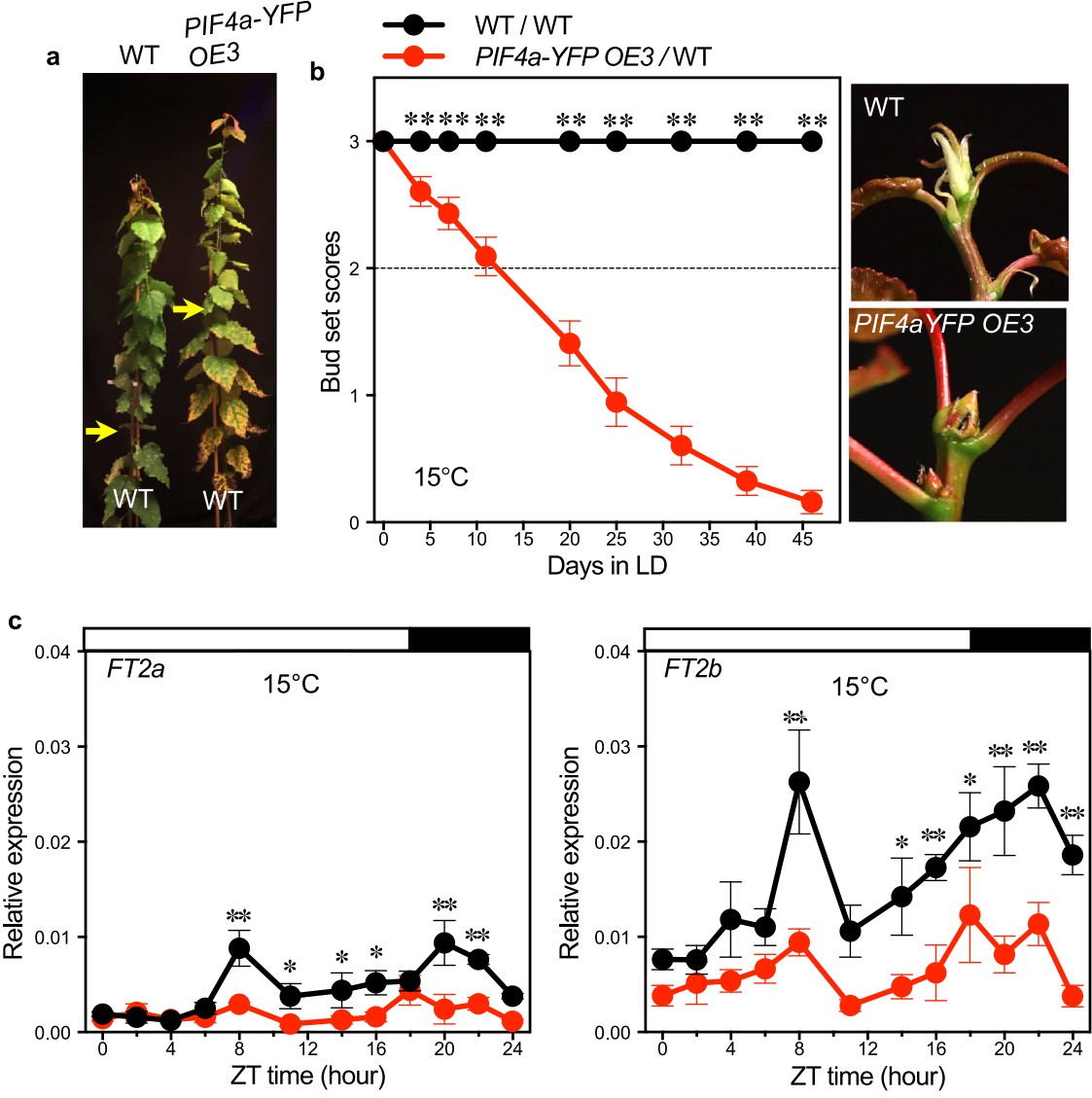

**Fig. 3 | *PIF4a* overexpression suppresses *FT2* transcription. a** Plants grafted with WT and *PIF4a-YFP OE3* scions onto WT rootstocks. The arrows indicate the grafting positions. **b** Bud set scores of WT and *PIF4a-YFP OE3* grafted plants in 15 °C LD. Shoot apices, 40 days after grafting, are shown in the right panel. The dotted line marked bud set score 2, which indicates the stage of growth cessation. **c** Diurnal expression of *FT2a* (left) and *FT2b* (right) in leaves of WT and *PIF4A-YFP OE3* scions grown for 21 days in 15 °C LD. Data are presented as mean ± SEM (*n* = 30 for **b**; *n* = 3 for **c**). The asterisks represent significance levels (*\*p* < 0.05, *\*\*p* < 0.01) based on Fisher's LSD test (two-sided) without multiple testing correction following ANOVA. See also Supplementary Fig. 3. Source data are provided as a Source Data file.

## PIF4 binds to a negative *cis*-element in the *FT2* genomic region

To investigate the mechanism for the suppressing role of PIF4 in regulating *FT2* transcription, we searched for potential PIF4 binding sites in the genomic region (29 kb) of the tandemly duplicated *P. tremula FT2* paralogs[16] (Fig. 4a), including G-box elements (CACGTG) and E-box elements (NACGTG, CACNTG)[31–35]. We identified two E-boxes in the *FT2a* and *FT2b* intergenic regions (Fig. 4a). We then performed yeast one-hybrid assays to investigate the PIF4-binding activity on the E-boxes from the *FT2* intergenic genomic DNA fragment, *proFT2$_{E-box}$* (1273 bp). The previously reported interaction between PtrSVL and a *PtrFT1* promoter fragment was used as a positive control (Fig. 4b)[36]. The data shows that PIF4a can bind to the *proFT2$_{E-box}$* fragment (Fig. 4b). To confirm in vivo binding of PIF4a to E-box elements, we conducted Cleavage Under Targets and Tagmentation (CUT&Tag)

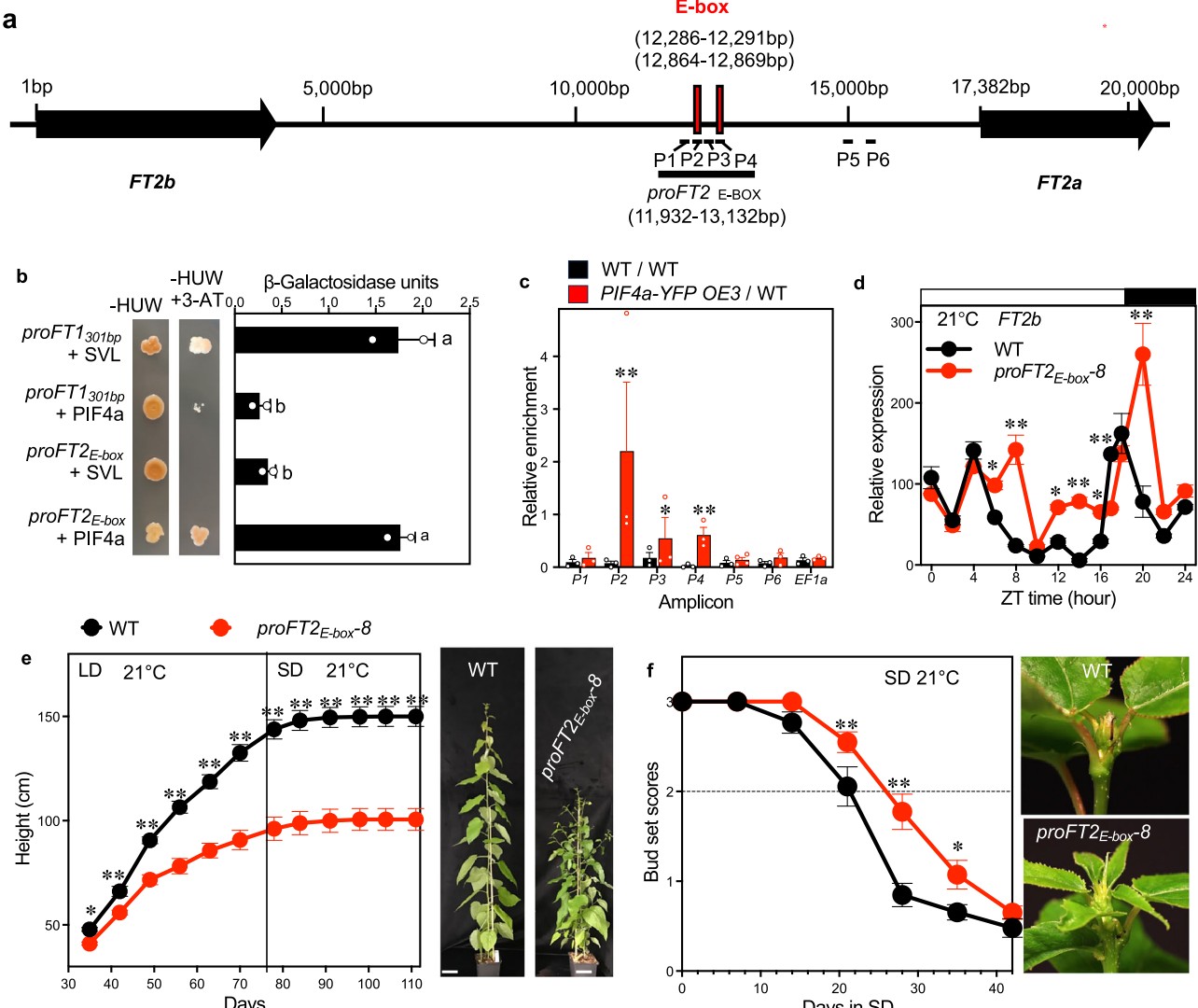

**Fig. 4 | PIF4a binds to a negative cis-element in the *FT2* genomic region.**
**a** Genomic organisation of *FT2* paralogs in *Populus tremula*. Black boxes indicate genomic regions of the two paralogs from the start to the stop codons. The red box shows the E-box elements (NACGTG, CACNTG). **b** Yeast one-hybrid assays of PIF4 binding to the E-box region. The indicated plasmid combinations were co-transformed into a yeast reporter strain, and interactions were assessed by growth on selective media and colourimetric assay for β-Galactosidase activity. 50 mM 3-amino-1,2,4-triazole (3-AT) was used to suppress autoactivation[84]. The combinations of *proFT1₃₀₁bp* and PtrSVL served as a positive control[22,36]. Negative controls are represented by the vector combinations of *proFT1₃₀₁bp* + PIF4a and *proFT2_E-box* + PtrSVL. **c** Relative enrichment of intergenic fragments after CUT&Tag quantified by qPCR. Values are normalised against the input DNA. The unrelated elongation factor 1 alpha (*EF1a, Potra2n6c14161*) served as a negative control. **d** Diurnal

expression of *FT2b* in leaves of the *proFT2_E-box-8* CRISPR plants grown for 40 days in 21 °C LD. Height growth (**e**) and bud set scores (**f**) of WT and the *proFT2_E-box-8* mutant. Plants were grown in LD at 21 °C for 77 days, then shifted to SD at 21 °C. Representative pictures of whole trees in LD and shoot apices in SD are shown in the right panel. Bars indicate a length of 10 cm. The dotted line marked bud set score 2 indicates the stage of growth cessation[26]. Data are presented as mean ± SEM ($n = 2$ for **b**, **d**; $n = 3$ for **c**; $n = 10$ for **e**, **f**). Fisher's LSD post hoc tests (two-sided, uncorrected for multiple comparisons) were performed following ANOVA to compare different plasmid combinations (**b**), various chromatin positions (**c**), and distinct time points (**d**, **e**, **f**). The statistically significant differences are indicated by non-matching letters in (**b**) ($p < 0.05$). Asterisks in (**c**, **d**, **e** and **f**) indicate the levels of statistical significance, * $p < 0.05$, ** $p < 0.01$. See also Supplementary Fig. 4 and Supplementary Table 1, 2. Source data are provided as a Source Data file.

assays[37,38] using leaves from *PIF4a-YFP OE3* grafted plants. qPCR analysis revealed significant enrichment at three fragments (P2, P3, P4) spanning the two E-box regions in *PIF4a-YFP OE3* plants compared to WT self-grafted controls (Fig. 4c). No enrichment was observed at control loci (P1, P5, P6, and *EF1α*) (Fig. 4c). Collectively, these findings demonstrate that PIF4a specifically associates with the *proFT2_E-box* region and likely mediates transcriptional repression through direct DNA binding.

Since the *proFT2_E-box* fragment is 8.2 kb downstream of *FT2b* and 4.2 kb upstream of *FT2a*, we wanted to know how this particular fragment affects the expression of the *FT2* genes in aspen trees. Hence,

we designed guide RNAs targeting two sites spanning the *proFT2_E-box* fragment to generate fragment-deleted mutants using CRISPR/Cas9 technology (Supplementary Fig. 4a). Possibly due to DNA self-repairing systems in the cell that caused difficulties in generating full fragment-deleted plants at both alleles simultaneously[39], we could only obtain heterozygous deletion lines (Supplementary Table 2. Supplementary Fig. 4). Intriguingly, even the heterozygous *proFT2_E-box* fragment deleted mutant showed pronounced expression peaks shifted for both *FT2a* and *FT2b* (Supplementary Fig. 4b, Fig. 4d). The *FT2b* expression peak at the end of the day was largely enhanced in the mutant (Fig. 4d). Consistent with that, the CRISPR/Cas9 mutant

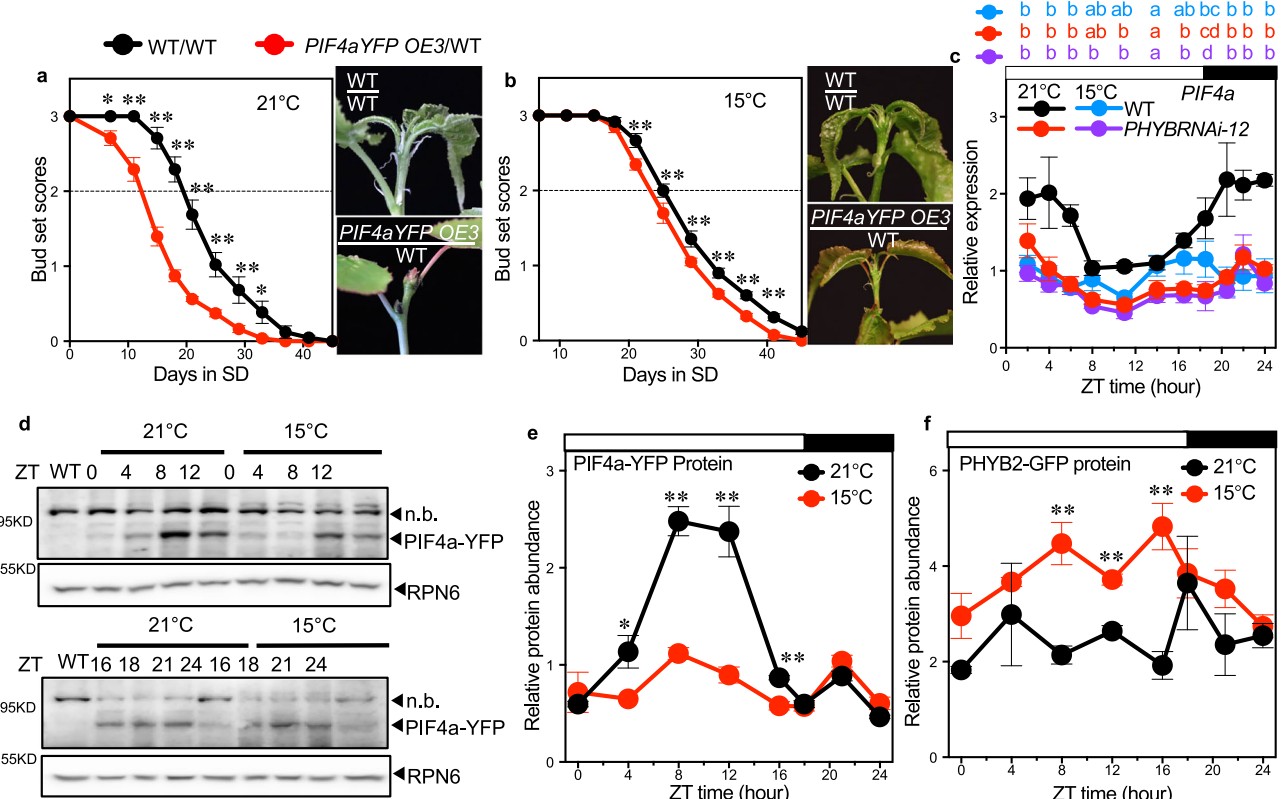

**Fig. 5 | Low temperature suppresses *PIF4s* function at both mRNA and protein levels.** Bud set scores of WT and *PIF4a-YFP OE3* grafted plants during SD-induced growth cessation at 21 °C (**a**) and 15 °C (**b**). Representative pictures of the shoot apices are shown in the right panel. The dotted line marked bud set score 2, which indicates the stage of growth cessation[26]. Pictures were taken when the bud set score of WT plants reached stage 2. **c** Diurnal relative gene expression of *PIF4a* in leaves of WT and *PHYBRNAi-12* trees grown for 40 days at 21 °C and 15 °C. **d** Western blot analysis of PIF4a-YFP fusion protein in *PIF4a-YFP OE3* plants at 21 °C and 15 °C using an anti-GFP antibody to detect PIF4a-YFP and an anti-RPN6 antibody as a loading control. Leaf samples were taken at ZT0, 4, 8, 12 (shown in the upper panel), and ZT16, 18, 21, and 24 (shown in the lower panel). *n.b.* indicates a non-specific band. **e** Quantification of PIF4a-YFP fusion protein from western blots. **f** Quantitative analysis of western blots of PHYB2-GFP fusion protein in *PHYB2-GFP OE6* soil-grown plants under LD conditions at 21 °C and 15 °C. Leaf samples were collected at ZT0, 4, 8, 12, 16, 18, 21, and 24. The original western blots are shown in Supplementary Fig. 5c. Data are presented as mean ± SEM (*n* = 10 for **a**, **b**; *n* = 3 for **c**, **e**, **f**). Fisher's LSD post hoc tests (two-sided, uncorrected for multiple comparisons) were performed at all time points following ANOVA. The statistically significant differences are indicated by non-matching letters in (**c**) ($p < 0.05$). Asterisks in **a**, **b**, **e**, **f** indicate the levels of statistical significance, *$p < 0.05$, **$p < 0.01$. See also Supplementary Fig. 5 and Supplementary Table 1. Source data are provided as a Source Data file.

displayed a bushy phenotype similar to that shown in *FT* over-expressing trees (Fig. 4e)[16,17]. Interestingly, these mutants showed delayed SD-induced growth cessation at both 21 °C and 15 °C (Fig. 4f, Supplementary Fig. 4c, d), suggesting that the *cis*-element is a general negative regulator regardless of temperature. Altogether, these results strongly suggest that PIF4 specifically responds to low temperatures and acts as a repressor interacting with an *FT2* genomic site that negatively regulates the transcription of both *FT2a* and *FT2b*.

**Low temperature reduces PIF4 abundance at both mRNA and protein levels**

As previously shown, low temperatures induce PHYB's transcriptional levels. We then wanted to investigate how lower temperature affects PIF4 function in aspen trees. In SDs, the earlier growth cessation phenotypes of *PHYBRNAi-12* plants at 15 °C were suppressed in the *PIF4RNAi* background (Fig. 2d, e). Consistent with that, *PIF4a-YFP OE3*(scion)/WT(rootstock) plants ceased growth seven days earlier in 21 °C SD but only two days earlier in 15 °C SD compared to WT self-grafted plants (Fig. 5a, b). Notably, while WT plants showed significantly suppressed height growth at 15 °C LD relative to 21 °C LD, soil-grown *PIF4a-YFP* overexpressing plants exhibited significantly increased growth at 15 °C compared to 21 °C (Supplementary Fig. 3c).

These results suggest that low temperatures can largely suppress PIF4's function.

In Arabidopsis, PIF4 is activated by elevated ambient temperature at both the transcriptional and post-transcriptional levels[40–42]. To understand how lower temperatures affect PIF4 in our trees, we first analysed the diurnal expression patterns of *PIF4a* at 21 °C and 15 °C. The *PIF4a* transcription levels were significantly downregulated in the *PHYBRNAi-12* trees, particularly at 21 °C (Fig. 5c), suggesting that *PHYB* promotes *PIF4* transcription, which is opposite to the findings in Arabidopsis[43]. Interestingly, the low temperature-induced suppression of *PIF4a* expression in WT plants was abolished in the *PHYBRNAi-12* background (Fig. 5c), suggesting that PHYB does not significantly affect PIF4's transcription at low temperatures.

Active PfrB promotes PIF4 protein degradation in the Arabidopsis[41,44–46]. Hence, we hypothesised that a higher level of *PHYB* at low temperatures could result in even lower levels of PIF4 protein accumulation in aspen trees. We, therefore, investigated the diurnal pattern of PIF4a-YFP fusion protein abundance in LD at 21 °C and 15 °C in the *PIF4a*-overexpressing plants. While at 21 °C the PIF4a-YFP fusion protein accumulated at its highest levels between ZT8 and ZT12, the levels were significantly reduced at 15 °C (Fig. 5d, e). At the same time, low temperatures did not significantly decrease the accumulation of

the *PIF4a-YFP* transcript (Supplementary Fig. 5a). Similar results were also observed from the in vitro-grown plants (Supplementary Fig. 5b). Furtermore, the PHYB2-GFP fusion protein accumulated to significantly higher levels in LD (ZT8-16) at lower temperatures, mirroring the observed changes in PIF4 protein abundance (Figs. 5f, S5c). Collectively, these results indicate that under permissive LD conditions at low temperatures, aspen trees elevate *PHYB* mRNA and protein levels to post-translationally suppress PIF4 activity, thereby preventing premature growth cessation.

### Low temperature enhances the PHYB2–PIF4 interaction to promote PIF4 degradation

To further investigate whether temperature affects light-triggered PIF4a protein degradation in trees, we conducted PIF4a protein stabilisation assays in a *35S::PIF4a-HA* transgenic aspen tree in response to red light treatment at different temperatures. Plants were first kept in the dark at 21 °C for 24 h and then subjected to 10 µmol/m²/s red light treatment either at 21 °C or 15 °C. As shown in Arabidopsis[41], red light triggered proteasome-mediated PIF4a-HA degradation, where PIF4a-HA protein abundance was significantly reduced after 30 min of red light treatment (Supplementary Fig. 6a). After 60 min, PIF4a-HA protein abundance was reduced by about half the amount at 15 °C compared to 21 °C (Fig. 6a), suggesting that low temperature enhances red-light-triggered PIF4a protein degradation.

To determine whether the protein abundance difference resulted from a temperature-dependent alteration in PIF4a polyubiquitination, we performed pull-down assays with a Tandem Ubiquitin Binding Entities (TUBEs) approach to detect polyubiquitinated PIFa-HA protein accumulation in response to temperature. Total ubiquitinated proteins from light/temperature-treated samples were purified using agarose beads coupled with TUBEs and then detected by western blotting using anti-HA and anti-ubiquitin antibodies. A set of closely migrating high molecular weight proteins was more abundant in post-30 min red light-treated plants, indicating a light-induced polyubiquitination of PIF4a-HA (Fig. 6b, Supplementary Fig. 6b). Polyubiquitinated PIF4a-HA was found to be substantially more abundant after 30 and 60 min of red light exposure at 15 °C relative to 21 °C. (Fig. 6b). This result indicates that low temperature enhances light-induced PIF4a-HA polyubiquitination.

We then wanted to know whether temperature could affect the potential PHYB2-PIF4 protein–protein interactions in these trees, as recently described in Arabidopsis, where increasing temperatures lead to an increased dissociation of the phyB-PIF interaction[47]. We generated *PHYB2* and *PIF4a* co-overexpressing plants, in which a *35S::PHYB2-GFP* construct was introduced into the *35S::PIF4a-HA* transgenic background line described above. We first characterised PIF4a-HA protein abundance with/without *PHYB2* co-overexpressed at two temperatures. Consistent with the red-light treatment, ectopically expressed *PHYB2* significantly reduced PIF4a-HA protein accumulation at 21 °C, significantly enhancing this reduction at 15 °C (Supplementary Fig. 6c). We then performed co-immunoprecipitation assays, in which PHYB2-GFP and PIF4a-HA protein were co-precipitated in the same complex (Fig. 6c). Interestingly, the PHYB2-GFP and PIF4a-HA protein interaction was much more enhanced at 15 °C than at 21 °C (Fig. 6c). TUBE assays showed that co-expression of PHYB2-GFP significantly increased PIF4a-HA polyubiquitination at 21 °C (Fig. 6d, see the blot of *long exp.*) and at 15 °C, this polyubiquitination was even more enhanced (Fig. 6d, see the blot of *short exp.*). Together, these results proved that low temperatures increase the PHYB-PIF4a protein interaction to enhance PIF4a protein degradation.

## Discussion

The effects of a cold climate on plant growth include both gradual influences and extremes[48–50]. The gradual influences constrain but allow the growth without severe freezing. Perennial plants must establish dormancy to survive the low-temperature extremes of winter[11,51,52]. Plants adapted to growth at lower temperatures (8 °C) during the growing season surprisingly showed almost the same monthly productivity as those growing at higher temperatures (28 °C)[23,50], suggesting that these plants have evolved successfully to cope with a colder environment. Some reports showed that cold-adapted plants could increase the number of mitochondria or the efficiency of respiratory machinery to keep up the growth[53,54]. However, the molecular mechanisms for how plants adapt to growth at lower temperatures are still largely unknown. Until recent years, some significant breakthroughs have been made in plant thermomorphogenesis signalling[1,3], which opened up new avenues for studying low-temperature growth. Several proteins with roles in temperature-sensing mechanisms in Arabidopsis have been reported, including *AtPHYB*[24,25], *AtPIF7*[55,56], and *AtELF3*[57]. However, the putative temperature sensing domains of *AtPIF7* and *AtELF3* are missing in their poplar orthologs[58]. We show here that *PHYB* is required for the low-temperature growth of poplar trees, suggesting that *PHYB* plays a significant role in temperature sensing in these trees.

*PHYB* has been shown to regulate plant morphology and physiology in response to environmental changes, including seed germination, shade avoidance, flowering, and biotic/abiotic stresses[59]. Apart from that, *PHYB* has also been implicated in regulating seasonal growth in trees[21]. In this study, we show that *PHYB* plays a crucial role in regulating aspen tree seasonal growth in colder climates. Aspen trees elevate the levels of both *PHYB* transcript and protein under low-temperature, permissive LD conditions (Fig. 5f, Supplementary Fig. 1a, S5c). In addition, lower temperatures are known to increase the levels of the active PfrB form of the protein[24,25], which would presumably lead to an even higher overall PHYB activity. The unexpected growth cessation observed in *PHYB*RNAi plants at 15 °C LD suggests that this increase in PHYB activity is crucial for maintaining vegetative growth at lower temperatures that would otherwise trigger a premature growth cessation. However, the temperature response in short-day (SD) conditions differs, especially in *PHYB2-GFP* overexpressing plants, which exhibited earlier SD-induced growth cessation at low temperatures (Fig. 1e, f). This finding points to substantial crosstalk between photoperiod and temperature signalling pathways. Notably, we observed significantly reduced PHYB2-GFP protein levels under low-temperature SD conditions (Supplementary Fig. 1j, k). Given that photoperiod is known to play a dominant role in regulating growth cessation[7,11,12], this result implies that short days can override the effects of low temperature on growth by suppressing PHYB2-GFP protein accumulation.

Although the mechanisms of phyB as a thermosensor have been well described biochemically and genetically[24,25], it is still unclear how phyB passes on the temperature signal to the downstream target genes. Jung et al., proposed that phyB could suppress target gene transcription by competing with PIF4, specifically at low temperatures[25]. Kumar et al. showed that the PIF4 protein was not affected by temperature in Arabidopsis[30]. However, our study shows that temperature not only regulates the abundance of PHYB and PIF4 in aspen trees but that low temperatures also significantly enhance the PHYB-PIF4 protein–protein interaction, further inducing proteasome-mediated PIF4 degradation (Fig. 6). These findings suggest possible evolutionary differences in the functional regulation of the phyB-PIF4 modules.

In Arabidopsis, phyB represses *FT* expression by destabilising its primary activator, the CONSTANS (CO) protein[60]. On the contrary, phyB in poplar promotes *FT2* expression (Fig. 1c, d)[21]. A single temperature decrease can facilitate Arabidopsis CO protein accumulation in the morning but does not result in an *FT* upregulation[61]. In contrast, *FT* was downregulated by low temperatures at dusk[61]. These studies suggested that other players downstream of phyB are involved in temperature-controlled *FT* regulation in Arabidopsis. Similar to

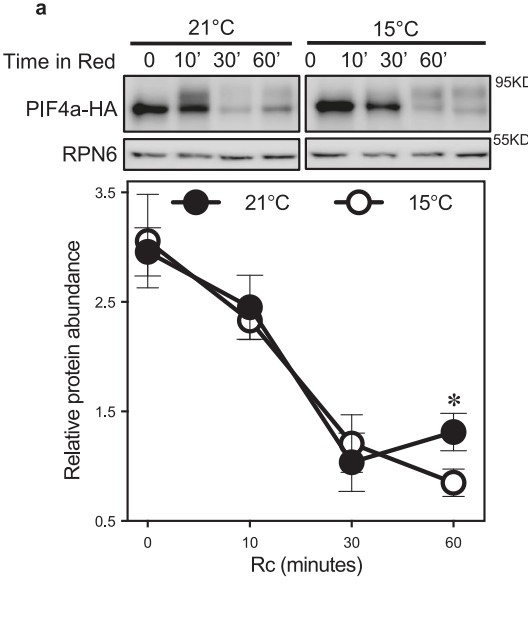

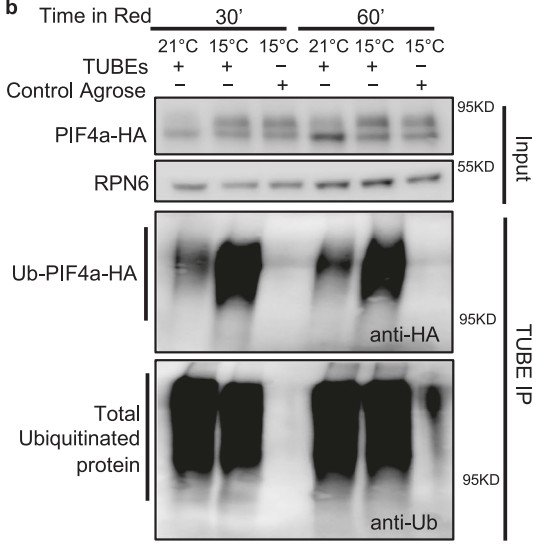

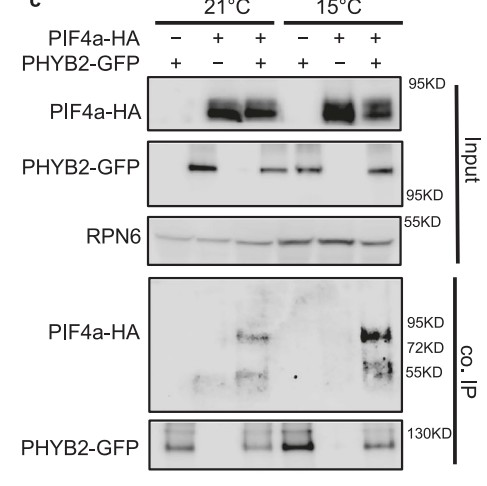

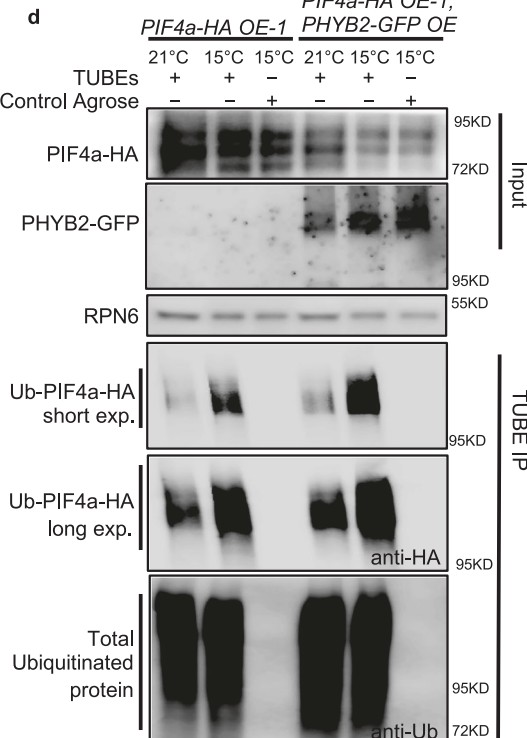

**Fig. 6 | Low temperature enhances the PHYB2-PIF4 interaction to promote PIF4 degradation. a** Western blot of PIF4a-HA fusion protein in *PIF4a-HA* over-expressing plants in response to red light at 21 °C and 15 °C using an anti-HA antibody to detect PIF4a-HA and an anti-RPN6 antibody as loading controls. 4-week-old plants were kept in the dark at 21 °C for 24 h and then transferred into two cabinets with 10 μmol m⁻² s⁻¹ red light at either 21 °C or 15 °C. Leave samples were taken at 0, 10, 30, and 60 min of red light treatment at two temperatures. The bottom panel shows the quantification of PIF4a-HA fusion protein by western blot. Data are represented as mean ± SEM, *n* = 3. The asterisks represent sig-nificance levels (* *p* < 0.05) based on Fisher's LSD test (two-sided) without mul-tiple testing correction following ANOVA. **b** TUBEs (tandem ubiquitin-binding entities) assays of ubiquitinated proteins in the samples from (**a**) after 30 and 60 min of red light treatment. Total ubiquitinated proteins from samples were immunoprecipitated with agarose-TUBE2, then analyzed by western blotting with

anti-HA antibodies for detection of PIF4-HA and anti-ubiquitin antibodies as loading controls. Control agarose that was not TUBEs-conjugated was used as a negative control. Anti-RPN6 antibodies were used as loading controls for input samples. **c** Co-immunoprecipitation assays of PHYB2 and PIF4a interaction in *35S::PIF4a-HA-1/35S::PHYB2-GFP* trees at 21 °C and 15 °C. Co-immunoprecipitated PIF4a-HA was detected by western blotting using anti-HA antibodies, after immunoprecipitation of PHYB2-GFP by GFP-trap beads. Precipitate probed with anti-GFP antibodies and input samples are shown as controls. **d** TUBEs assays of ubiquitinated proteins in *35S::PIF4a-HA-1* and *35S::PIF4a-HA-1/35S::PHYB2-GFP* trees at two temperatures. Leaf samples were taken at ZT8 of 6-week-old LD grown trees. One short-time exposed image and one long-time exposed image are shown from the same blot. See also Supplementary Fig. 6. Source data are pro-vided as a Source Data file.

Arabidopsis, we also observed that the two poplar *FT2* paralogs were significantly downregulated at low temperatures (Supplementary Fig. 7a). Moreover, CO plays a minor role in the poplar tree photoperiod response[17,62]. These findings suggest that in both Arabidopsis and poplar, CO is not a major player in regulating *FT* transcription in response to lower temperatures.

Photoperiod serves as the primary regulator of autumn growth cessation in poplars, though temperature also plays a significant modulatory role. Recent evidence demonstrates that exposure to 12 °C can induce premature growth cessation in aspen under LD conditions[5]. While the molecular basis of photoperiodic control is well characterised in *Populus*, the mechanisms governing temperature responses remain elusive. Notably, although low temperatures strongly suppress *FT2* expression in poplar, vegetative growth persists at 15 °C, implying the existence of a critical *FT2* expression threshold required for growth maintenance, as previously suggested from studies of trees with different levels of *FT* downregulation[17]. Our work reveals that the PHYB-PIF4 module specifically and critically sustains *FT* transcription to promote vegetative growth under these otherwise non-permissive temperature conditions.

Consistent with findings in Arabidopsis where *FT*, *PIF4* and *PHYB* show overlapping expression in minor leaf veins[30,63], we observe similar co-expression patterns in *Populus* leaves. Our whole-leaf expression analysis in aspen reveals highly coordinated expression of *PHYB*, *PIF4* and *FT2*, strongly indicating their functional interaction in vegetative tissues. This observation is further supported by RT-PCR analysis, which clearly demonstrates co-expression of these genes in tissues containing minor veins (Supplementary Fig. 7b), recapitulating the Arabidopsis expression pattern.

Arabidopsis *FT*-regulating transcription factor binding sites are well-characterised[64]. Similar to our findings, no G-box elements have been identified in the *FT* upstream promoter region. Current literature reports only one functional G-box element located 1.5 kb downstream of the Arabidopsis *FT* gene which serves as a confirmed PIF4 binding site[65]. Kumar et al. demonstrated that PIF4 binds to cis-elements near the *FT* start codon in Arabidopsis, where multiple E-boxes are present[30,65]. Other thermoresponsive regulators (e.g., SVP[66], FLM[67], and FLC[68]) also target this region. Our findings proved that PIF4 binds to an intergenic region of two *FT2* paralogs in poplar trees. This result suggested that PIF4-mediated *FT* regulation under lower temperatures appears to be a *Populus*-specific adaptation, likely reflecting its critical role in regulating growth cessation and bud set.

PIF4 in Arabidopsis acts as a central signalling hub in the temperature response, integrating various upstream thermo-signals to reshape thermomorphogenesis[1,28–30,42]. The binding of PIF4 to the promoter of Arabidopsis *FT* increases at elevated temperatures, resulting in an upregulated *FT* expression promoting flowering[30]. However, we show that PIF4 suppresses *FT2* transcription in poplar trees (Figs. 2f, 3c). As seen in Arabidopsis, due to the functional redundancy of multiple PIFs, single loss-of-function mutants of AtPIFs do not show any obvious growth defects at normal growth conditions[46,69]. We hypothesise that the same redundancy exists in aspen trees that harbours seven PIFs[21]. Ding et al., also showed the same suppressing function of PIF8 on *FT2* expression[21]. These findings all indicate a general negative role of aspen trees' PIFs on *FT* expression. The high-order mutants of aspen PIFs would potentially further prove the conclusion. Interestingly, PIF8 was found, in contrast to PIF4, to play a significant role during the SD-induced growth cessation at 21 °C[21]. Taken together, these results suggest divergent functions of different PIFs in SD-induced growth cessation, in which PIF8 is a major player and is joined by PIF4, specifically when the temperature drops in autumn, to fine-tune the response (Supplementary Fig. 8). The downregulation of *FT2b* in the PIF4a binding site deletion mutant suggests that this regulatory region could be involved in recruiting more partners into a complex suppressing *FT2* transcription. The discovery of

four additional potential PIF4-binding E-box elements in the upstream region of *FT2b* provides further support for a regulatory role of PIF4a in controlling *FT2b* expression (Supplementary Fig. 4a). These results suggest an evolutionary divergence of PIF4's functional mode between the two model species, *Arabidopsis* and *Populus*.

There are two more pieces of evidence to support this functional divergence. First, diurnal expression analysis shows that PIF4 peaks during the evening in poplar trees (Fig. 5c) while it peaks during the day in Arabidopsis[70,71]. A previous study showed that *PIF4* transcript levels in Arabidopsis were controlled by the internal circadian clock[72]. Unlike the day-length-dependent induction of *FT* by CO protein in the Arabidopsis, some studies showed that night length plays a determining role in *FT* transcription in poplar trees[22,73]. These data suggest that PIF4 controls *FT* transcription differently in these species, resulting from an evolutionary divergence of circadian oscillators and diurnal environmental responses in trees. Second, in Arabidopsis, low temperatures suppressed PIF4's binding activity without affecting its protein abundance[30]. However, in poplar trees, PIF4 protein accumulation was significantly decreased by low temperatures (Fig. 5e, Supplementary Fig. 5b). All these findings suggest a different mechanism for regulating *PIF4* function in trees by internal and external signals and that this difference is important for the tree's ability to cease growth and set bud at the correct time, taking both photoperiod and temperature signals into account.

## Methods

### Plant material and growth conditions

Hybrid aspen (*Populus tremula x tremuloides*) clone T89 was used as WT control, and genetic modifications were conducted in this background, except that the *Populus tremula* clone SwAsp15 was used as background to generate *FT2* promoter deletion mutants. The hybrid aspen was originally obtained from central Europe, where the critical day length for its growth is 15.5 h, and the optimal growth temperature is 20–22 °C[74]. The SwAsp15 clone was collected from southern Sweden. Plants were cultivated on ½ Murashige and Skoog medium under sterile conditions for four weeks or until they had rooted (max. 8 weeks). After being transferred to soil, plants were grown in growth chambers and fertilised every second week (10 mL NPK-Rika S/plant). All growth conditions in this study were set up to LD (18 h light/6 h dark) or SD (14 h light/10 h dark) unless specified. Temperatures were set to 21 °C (optimal temperature) or 15 °C (cold environment). Illumination was from 'Powerstar' lamps (HQI-T 400 W/D BT E40, Osram, Germany), giving an R/FR ratio of 2.9 and a light intensity of 150–200 µmol m$^{-2}$ s$^{-1}$. To induce growth cessation, trees were moved to SD (14 h light/10 h dark) for up to 12 weeks, and fertilisation was stopped. In SD and LD, previously published bud scores were used to assess the effects on the bud set[26].

### Grafting experiments

Grafting was conducted as previously described[16]. In brief, scions of soil-grown plants were grafted onto rootstocks after five weeks in growth chambers (18 h light/6 h dark, 21 °C). Self-grafted WT scions onto WT rootstocks were used as control plants. More than twelve grafted plants for each of *PIF4a-YFP OE3*/WT and WT/WT combinations were investigated.

### Generation of CRISPR/Cas9 mutants

The CRISPR/Cas9 construct was made following the protocol described previously[16]. GreenGate entry and destination vectors were acquired from Addgene[75]. Potential sgRNAs for target genes were identified with E-CRISP. They were introduced into entry vectors by site-directed mutagenesis PCR. The final vector (containing promoter, Cas9 CDS, terminator, two sgRNAs and resistance cassette) was assembled by GreenGate reaction (150 ng of each component, 1.5 µL FastDigest buffer, 1.5 µL of 10 mM ATP, 1 µL 30U/µL T4 ligase and 1 µL

Eco31l in a 15 μL reaction) in 50 cycles of 5 min restriction/ligation at 37 °C and 16 °C, respectively, followed by 5 min 50 °C and 5 min 80 °C. All reagents were purchased from Thermo Scientific.

Vectors with the two guide RNAs (Supplementary Table 1) were transformed into *Populus tremula* clone SwAsp15 using a standard protocol[76]. Twenty-five transgenic lines were screened for target gene deletions using PCR (Supplementary Tables 1,2). Three independent transformants with biallelic modifications were initially characterised for *FT2* expression before selecting one line for deeper analysis.

### Generation of Y1H regulatory DNA fragments

Upstream regulatory DNA fragments used as bait in the Y1H assay were synthesised and cloned into a pUC57-BsaI-free vector (GenScript). The synthesised DNA fragment was recombined into pDONR P4-P1R (Life Technologies) and subsequently subcloned into the Y1H destination vectors pMW#2 and pMW#3 containing *HIS3* and *LacZ*, respectively[77], using LR Clonase (Life Technologies). Colony PCR was performed using M13F and HIS293RV primers for pMW#2 and 1H1FW and Lac592RV for pMW#3, while DNA sequencing to verify insert identifies uses M13F (for pMW#2 clones) and 1H1FW (for pMW#3 clones). Primers are listed in Supplemental Table 1.

The resulting DNA fragment baits were integrated into the genome of yeast strain YM4271 (*MATa, ura3-52, his3-200, lys2-801, ade2-101, ade5, trp1-901, leu2-3, 112, tyr1-501,gal4Δ gal80Δ, ade5::hisG*) (Clontech, Mountain View, CA) and selected by growth in dropout medium lacking histidine and uracil (SC-His-Ura). Colony PCR confirmed integration with primer pairs (Supplementary Table 1). Each yeast strain was tested for autoactivation by growth in SC-His-Ura medium containing increasing concentrations (10, 20, 40, 50, 80 mM) of 3-amino-1,2,4-triazole (3-AT; Sigma Life Sciences, St. Louis, MO), as previously described[78]. The coding DNA sequence of TFs was cloned into pENTR/D cloning (Life Techologies) and transferred into pDEST22 via Gateway recombination-based cloning to create a GAL4-activation domain fusion Y1H prey vector (Life Techologies).

For yeast one-hybrid screens, positive interactions were identified by both selection of colonies on SC-Histidine-Uracil-Tryptophan (SC-UHW) with different concentrations of 3-AT[79], and colourimetric assay for β-Galactosidase activity[80].

### RNA quantification

Total RNA extractions were conducted as previously described[16]. In brief, 100 μl ground poplar leaf powders were used for total RNA extraction with Cetyltrimethylammonium bromide (CTAB) buffer[81], including 2% CTAB, 100 mM Tris-HCl pH 8.0, 25 mM EDTA, 2 M NaCl, and 2% PVP4000. The samples were incubated at 65 °C for one minute, then extracted two times with a qual volume of chloroform-isoamyl alcohol (24:1). Nucleic acids were precipitated at 4 °C for 1 h with 1/3 volumes 8 M LiCl. Precipitates were collected by centrifugation at 13,000 × g for 20 min and purified with an RNeasy kit (Qiagen). DNase treatment was performed on-column (Qiagen).

cDNA was synthesised with an iScript™ cDNA Synthesis kit (Biorad) according to the manufacturer's instructions. Quantitative real-time PCR was conducted in a CFX384 Touch Real-Time PCR Detection System (Biorad) with SsoAdvanced SYBR mix (Biorad). Relative expression levels were obtained using the $2^{-\circ\Delta\Delta Cq}$ method[82]. Housekeeping genes *YLS8* (*Potra003266g21171*) and *eIF5A* (*Potra001829g14736*) were determined by GeNorm[83]. A complete list of primers used in real-time PCR analysis is presented in Supplemental Table 1.

### Protein extraction and immunoblot

Protein extraction was carried out as previously described[41]. In brief, total protein was extracted from 100 μl ground poplar leaf powder in 2× SDS loading buffer, then loaded on an 8% SDS-PAGE gel and blotted onto an Immobilon-P PVDF transfer membrane (Millipore). The resulting immunoblot was probed with an anti-GFP polyclonal antibody (PAGB1, Proteintech) to detect PIF4a-YFP and phyB2-GFP protein and the anti-HA-POD monoclonal antibody (Roche) to detect PIF4a-HA protein. Band signals were visualised by the SuperSignal Western Blotting system (Thermo Scientific). The intensities of Western blot band signals were collected from the Azure 600 Imaging System (Azure) and were measured using Image J. Quantification was performed with three biological replicates for each time point using anti-RPN6 (26S proteasome non-ATPase regulatory subunit, Agrisera) or anti-tubulin (Sigma) as loading controls.

### Co-immunoprecipitation assay

Leave samples were harvested from LD-grown trees at ZT8 of both *PIF4a-HA OE-1* and *PIF4a-HA OE-1; phyB2-GFP*. Total protein was isolated with extraction buffer (EB) containing 25 mM Tris-HCl pH 7.8, 10 mM MgCl₂, 150 mM NaCl, 200 mM PMSF, 1 μg/ml aprotinin, 0.2% NP40, 40 μM MG132, and protease inhibitors (Protease Inhibitor Cocktail for plant cells, Sigma). For co-immunoprecipitation assays, 500 μg of total proteins were incubated for 3 h with 20 μl of GFP-Trap® Magnetic Agarose beads (Proteintech) under 10 μmol/m²/s red light at either 21 °C or 15 °C. Immunocomplexes were washed three times with EB buffer and eluted by boiling in 40 μl of 2x Laemmli sample buffer. Proteins were then resolved by SDS-PAGE and blotted to the PVDF transfer membrane (Immobilon-P, Millipore). The epitope-tagged proteins were probed with anti-HA-peroxidase (3F10, Roche) or anti-GFP antibodies (PABG1 Proteintech) and detected with the SuperSignal West Pico Chemiluminescent Substrate. An anti-RPN6 antibody (26S proteasome non-ATPase regulatory subunit, Agrisera) was used as loading control.

### TUBEs analysis

The Immunoprecipitation of ubiquitinated proteins from *PIF4a-HA OE-1* and *PIF4a-HA OE-1; phyB2-GFP* using Tandem Ubiquitin Binding Entities (TUBEs) agarose (tebu-bio, Le Perray-en-Yvelines, France) was performed as previously described with slight modification[41]. Proteins were extracted with a buffer containing 100 mM MOPS, pH7.6, 150 mM NaCl, 0.1% NP40, 1% Triton X-100, 0.1% SDS, 20 mM Iodoacetamide, 1 mM PMSF, 2 μg/l aprotinin, 40 μM MG132, 5 μM PR-619, 1 mM 1,10-Phenanthroline, and 2X Complete protease inhibitor Cocktail and PhosStop cocktail (Roche). 30 μl Agarose-TUBE2 was incubated with 5 mg total protein from each sample for 6 h at 4 °C. The agarose beads were washed with extraction buffer four times and eluted with 2x laemli loading buffer, then subjected to Western blot analysis with the 16B12 anti-HA-POD antibodies (Roche) for detection of PIF4-HA and anti-ubiquitin antibodies (sc-8017, Santa Cruz Biotechnology) for detection of ubiquitinated protein.

### CUT&Tag analysis

PIF4a-YFP OE3/WT and WT/WT grafted plants were grown under LD conditions. For each genotype, we collected one fully expanded leaf from three biological replicates at ZT8. Leaves were cross-linked in 10 mM sodium phosphate buffer (pH 7) containing 50 mM NaCl, 100 mM sucrose, and 1% formaldehyde, with vacuum infiltration performed twice for 10 min each. The cross-linking reaction was quenched by adding glycine to a final concentration of 125 mM. Samples were flash-frozen in liquid nitrogen and ground to a fine powder. Nuclei were extracted using nuclei isolation buffer (10 mM sodium phosphate pH 7, 100 mM NaCl, 10 mM β-mercaptoethanol, 11.5% 2-Methyl-2,4-Pentanediol) supplemented with a 1× protease inhibitor cocktail (Complete, EDTA-free).

CUT&Tag was performed essentially as described by Kaya-Okur et al.[37,38], with the following modifications. Instead of Concanavalin A-coated magnetic beads, we used gentle centrifugation (1000 × g) with a swing-out rotor to pellet nuclei during washing steps. Nuclei

were incubated with a primary antibody (rabbit polyclonal anti-GFP; Abcam ab290) overnight at 4 °C with constant agitation. Secondary antibody incubation (donkey anti-rabbit; Agrisera AS10 1014) was performed for 90 min at room temperature. Following chromatin tagmentation with pAG-Tn5, DNA was extracted via phenol:chloroform extraction. Tagmented DNA fragments were then amplified using i5 and i7 primers with NEB Next High-Fidelity 2× Master Mix (M0541). PCR products were purified using AMPure XP beads (Omega M1378) as described in Kaya-Okur et al.[38].

The quantification of targeted enriched DNA was performed using SsoAdvanced Universal SYBR Green Supermix (Bio-Rad) on an iQ5 thermal cycler (Bio-Rad). Wild-type samples without tagged protein served as negative controls. All primer sequences used for genomic fragment amplification are listed in Supplementary Table 1.

### Statistics and reproducibility

All phenotypic analyses were repeated at least three times, and all RNA/protein assays were performed at least twice, with consistent results across experiments. No statistical method was used to predetermine sample size. No data were excluded from the analyses. The Investigators were not blinded to allocation during experiments and outcome assessment. Statistical analyses were performed using GraphPad Prime 10 software (Boston, Massachusetts USA, www.graphpad.com).

### Reporting summary

Further information on research design is available in the Nature Portfolio Reporting Summary linked to this article.

## Data availability

All data supporting the findings of this study are available with this paper and its Supplementary Information files and in the Source data file. Gene IDs and accesion numbers of this study are *PHYB1* (*Potra2n8c17574*), *PHYB2* (*Potra2n10c21137*), *PIF4a* (*Potra2n2c5930*), *PIF4b* (*Potra2n5c10998*), *FT2a* (*Potra2n10c20842*), *FT2b* (*Potra2n10c 20839*). All the gene accession numbers were obtained from the website https://plantgenie.org/. All unique/stable reagents generated in this study are available from the lead contact with a completed Materials Transfer Agreement. Source data are provided with this paper.

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

## Acknowledgements

This work was supported by the Swedish Research Council and the Knut and Alice Wallenberg Foundation to O.N.

## Author contributions

B.Z. and O.N. designed the research. B.Z., K.C.L, L.G.R., J.H.D. and A.M. performed the experiments and analysed the data. B.Z. and O.N. wrote the manuscript. All authors reviewed and approved its final version.

## Funding

## Competing interests

The authors declare no competing interests.
