## [Transparent Peer review · Nature Communications]

Phytochrome B and Phytochrome-Interacting-Factor4 Modulate Tree Seasonal Growth in Cold Environments

Corresponding Author: Professor Ove Nilsson

Version 0:

Reviewer comments:

Reviewer #1

(Remarks to the Author)

This manuscript describes how perennial plants, such as trees, adapt to varying environmental temperatures to optimize the seasonal growth and regulate growth cessation. Trees balance growth cessation and bud set to survive freezing conditions while maximizing growth during the growing season. Zhang et al., reveal that the PHYB-PIF4 module is crucial for maintaining growth in cold environments during growing seasons in poplar trees. High levels of PHYB induced under low temperatures positively regulate the expression of the FT2, preventing early growth cessation under LD. PIF4, a known downstream target of PHYB, reduces FT2 expression under low-temperature conditions. Protein-protein interaction between PHYB and PIF4 promotes the proteasome-mediated degradation of PIF4 via polyubiquitination, a process further enhanced at low temperature, thereby mitigating PIF4's negative effects on FT2 expression. Collectively, this study suggests that perennial plants have evolutionarily developed distinct mechanisms integrating light and temperature signals to optimize seasonal growth in response to environmental changes, thereby advancing our understanding of plant adaptation mechanisms to seasonal environmental changes. Below are some comments that need the authors to address.

Major comments:

Major #1. The authors mentioned that increment of active PfrB at lower temperatures might be critical to prevent the early growth cessation, on lines 154-155. In Fig. 1E and F, the PHYB2-GFP OE6 displayed delayed growth cessation compared to WT at 21°C. However, at lower temperature (15°C), growth cessation occurred earlier than at 21°C in PHYB2-GFP OE6, which tends to contrast with the differences in WT. How can the hypothesis that increased levels of active PfrB and its transcription at low temperature compromise the delayed growth cessation explains these phenomena? Would it be reasonable to consider that active PfrB increases alongside the transcriptional levels of it under low temperature conditions? The authors should provide data demonstrating the status of PHYB proteins at different temperatures.

Major #2. The authors claimed that PIF4 would directly suppress the expression levels of FT2 genes at low temperature via E-box motifs in the intergenic region between FT2b and FT2a (Fig 4). While proFT2E-box-8 deletion mutants result in significant reduction of expression levels of FT2a and FT2b, FT2b appears to be more influenced by PHYB and PIF4 (Fig. 1C and D, Fig. 2C and F, Fig. S2C and D, Fig. 3C) suggesting that the promoter region of FT2b might also play a crucial role in its regulation by PHYB and PIF4. Therefore, the authors should search whether there are additional potential PIF4 binding motifs upstream of FT2b, which could further support the regulatory mechanisms of FT2b.

Major #3. As shown in Fig. 5, the diurnal protein amount of PIF4 peaks during the daytime (ZT8-ZT12) at 21°C but significantly decreases under low temperature condition. This reduction likely impacts the PIF4-dependent regulation of FT2b expression (Fig. 2C and F, Fig. 3C, Fig. 4C). The direct interaction between PHYB and PIF4 is crucial for regulating PIF4 protein stability via ubiquitination-mediated proteasomal degradation (Fig. 6), which is further enhanced at low temperature condition. Consequently, the reduction in PIF4 protein amount during ZT8-ZT12 at low temperature should be closely affected by the status of active PfrB. To further strengthen the evidence for the role of the PHYB-PIF4 module in the diurnal regulation of FT2 transcription, the diurnal protein status of PHYB under different light conditions and temperatures should be examined.

Major #4. It has been well-documented that FT2 is predominantly expressed in leaf and reproductive buds during the growing season (Hsu et al., 2011). This suggests that PIF4, which functions as a direct upstream regulator of FT2 transcription, and PHYB, which modulates PIF4 protein stability through direct protein-protein interaction, may exhibit

overlapping spatial expression patterns with FT2. Therefore, the authors should investigate the spatial expression patterns of PHYB and PIF4 in poplar by generating appropriate reporter lines or through in situ hybridization.

Major #5. The authors mentioned 'significantly' on line 192, 195, 198, and 211 to describe the effect of PHYB or PIF4 on the regulation of growth cessation. However, the statistical significance of the data has not been confirmed. The authors should ensure that all claims of statistical significance are supported by appropriate analyses and validation (Fig. 1A, B, E, and F, Fig. S1B-G, Fig. 2A, B, D, and E, Fig. S2A and B, Fig. 3B, Fig. S3C, Fig. 4D and E, Fig. S4C and D, Fig. 5A and B, Fig. S6A). Additionally, in many cases, the figure legends do not specify the statistical methods used to analyze the data (Fig. 4B, Fig. S4B, Fig. S5B, Fig. 6A, Fig. S6C). Providing this information is crucial for ensuring transparency and reproducibility. The authors should include details on the statistical tests performed.

Minor comments:

1. The figure legend for Fig. 4D should clearly specify the plant growth conditions.
2. Line 347-349. The authors should support this claim with appropriate statistical analyses and rephrase it in a scientifically precise manner.
3. Line 519-520. Please provide the relevant data or include a reference to support this statement.

Reviewer #2

(Remarks to the Author)

The manuscript by Zhang et al., concerning the role PIF4 plays in the interplay between temperature and photoperiod in poplar trees provides very nice evidence for the divergent role of PIF4 across species. The interesting outcome of this manuscript is that it shows that PIF4 suppress FT2 transcription in poplar trees unlike Arabidopsis where PIF4 enhance the expression of FT. The authors provide supportive data that shows the role of Phyb plays in modulating the response to cold temperatures. The authors nicely demonstrate that FT2 promoter contains two E-boxes that can act as a binding site for PIF4 and attempted to CRISPR both sites, though unfortunately could only create heterozygous deletion lines though those lines still show some phenotype.

In general the manuscript is very nice and provides new evidence for the divergent role of PIF4 and PhyB in different plants. One major comment that I have is the E-box discovery. In general a discovery of this importance that can explain why PIF4 behaves different in poplar than arabidopsis could be more focused on. While indeed the yeast 1 hybrid show that PIF4 can bind the E-box, this is not definite and a ChIP and/or EMSA can be performed to show that PIF4 can indeed bind these regions in planta. This should be fairly straight forward especially as the authors already generated the PIF4-YFP OX lines.

Do the authors know if such Cis element also exist in Arabdiopsis plants that are adapted to cold temperature? or is this a specific adaptation for poplar?

Also as PIF4 is classified as a warm temperature signaling protein in Arabdiopsis, does it play any role in warm temperature in poplar?

A minor comment is in the way figures are mentioned in text, its better to sperate the letters such as Fig1A,B (See line 133 for example)

Version 1:

Reviewer comments:

Reviewer #1

(Remarks to the Author)

The authors have addressed most of my concerns, which I appreciate. However, one important issue remains unresolved and should be corrected to strengthen the conclusion of this manuscript.

While the authors present Western blot results in Figure S1J to support protein-level changes, the quality of the blot image is questionable. The bands appear inconsistent, especially for the loading control (RPNP6), making it difficult to assess the reliability of normalization and quantification. I strongly recommend providing a higher-quality blot with clearly consistent loading control signals to strengthen the reliability of the data.

Responses to comments

We sincerely appreciate the reviewers' thoughtful comments and valuable suggestions. In response, we have carefully addressed all points raised in the review process and incorporated detailed discussions throughout the manuscript. We have also provided additional experimental data as suggested by the reviewers, which we believe has significantly strengthened the quality and impact of our study.

Reviewer #1 (Remarks to the Author):

Major comments:

Major #1. The authors mentioned that an increment of active PfrB at lower temperatures might be critical to prevent the early growth cessation, on lines 154-155. In Fig. 1E and F, the PHYB2-GFP OE6 displayed delayed growth cessation compared to WT at 21°C. However, at lower temperature (15°C), growth cessation occurred earlier than at 21°C in PHYB2-GFP OE6, which tends to contrast with the differences in WT. How can the hypothesis that increased levels of active PfrB and its transcription at low temperature compromise the delayed growth cessation explains these phenomena? Would it be reasonable to consider that active PfrB increases alongside the transcriptional levels of it under low temperature conditions? The authors should provide data demonstrating the status of PHYB proteins at different temperatures.

This is indeed an intriguing observation. The primary biological role of increasing PfrB at low temperatures appears to be to prevent premature growth cessation under otherwise permissive long-day (LD) conditions. The elevated *PHYB* transcription levels observed at low temperatures in LD align well with our hypothesis. Since the temperature-dependent PrB-to-PfrB conversion mechanism is well-documented in plants (refs 24, 25), the presence and activity of PfrB protein is largely determined by the overall protein levels. We have now provided new protein data showing higher PHYB2-GFP protein abundance in low-temperature LD conditions (Figure 5F), further supporting our hypothesis.

However, the temperature response in short-day (SD) conditions differs, particularly in *PHYB2-GFP* overexpressing plants, suggesting significant crosstalk between photoperiod and temperature signalling pathways. As the reviewer suggested, to better understand the behaviour of *PHYB2-GFP* overexpressing plants, we analyzed PHYB2-GFP protein abundance in SD conditions at two temperatures. Interestingly, we observed significantly lower PHYB2-GFP protein levels in low-temperature SD conditions (soil-grown diurnal samples in Figure S1J and *in vitro*-grown samples in Figure S1K), consistent with the observed earlier growth cessation. Given that photoperiod is known to play a dominant role in regulating growth cessation (refs 7, 11,

12,16,17), this result implies that short days through the photoperiod pathway can override the effects of low temperature on growth by suppressing PHYB2-GFP protein accumulation. In other words, when the days are getting short enough during autumn there is no longer a need to prevent a premature growth cessation triggered by lower temperatures. With the newly added data, we have also discussed this point in the manuscript; please see lines 536-543.

Major #2. The authors claimed that PIF4 would directly suppress the expression levels of FT2 genes at low temperature via E-box motifs in the intergenic region between FT2b and FT2a (Fig 4). While proFT2E-box-8 deletion mutants result in significant reduction of expression levels of FT2a and FT2b, FT2b appears to be more influenced by PHYB and PIF4 (Fig. 1C and D, Fig. 2C and F, Fig. S2C and D, Fig. 3C) suggesting that the promoter region of FT2b might also play a crucial role in its regulation by PHYB and PIF4. Therefore, the authors should search whether there are additional potential PIF4 binding motifs upstream of FT2b, which could further support the regulatory mechanisms of FT2b.

Thanks for this suggestion. We have now performed an analysis of the *FT2b* promoter, searching for both G-box and E-box elements that PIF4 could potentially bind to. We examined the region up to -5 kb upstream of *FT2b*. While we did not identify any G-box elements, we did discover four additional E-box elements, consistent with those found in the intergenic region between *FT2b* and *FT2a*. We have also demonstrated binding of PIF4 to this new region (Figure S4E). As the reviewer noted, this finding further supports the regulatory role of PIF4 in controlling *FT2b*. We have updated Figure S4A to reflect this and included it in the discussion, lines 619-622.

Major #3. As shown in Fig. 5, the diurnal protein amount of PIF4 peaks during the daytime (ZT8-ZT12) at 21°C but significantly decreases under low temperature condition. This reduction likely impacts the PIF4-dependent regulation of FT2b expression (Fig. 2C and F, Fig. 3C, Fig. 4C). The direct interaction between PHYB and PIF4 is crucial for regulating PIF4 protein stability via ubiquitination-mediated proteasomal degradation (Fig. 6), which is further enhanced at low temperature condition. Consequently, the reduction in PIF4 protein amount during ZT8-ZT12 at low temperature should be closely affected by the status of active PfrB. To further strengthen the evidence for the role of the PHYB-PIF4 module in the diurnal regulation of FT2 transcription, the diurnal protein status of PHYB under different light conditions and temperatures should be examined.

Thanks for this helpful suggestion. We have now conducted a diurnal analysis of PHYB2-GFP protein levels under different photoperiods (LD and SD) and temperatures (21°C vs 15°C). The results showing temperature-dependent diurnal fluctuations of PHYB2-GFP under long-day conditions have been included in Figure 5F and Figure S5C. Notably, we observed a significant increase in PHYB2-GFP protein levels between ZT8-

16 at the lower temperature. This finding aligns well with our observation of reduced PIF4a-YFP protein accumulation (Figure 5E). We have incorporated these results in the text (lines 393-397).

We have also included the diurnal protein dynamics of PHYB2-GFP under short-day (SD) conditions at both temperatures in Figure S1J. The text (lines 536-543) and our response to Major #1 discuss the key differences observed between the LD and SD photoperiods.

Major #4. It has been well-documented that FT2 is predominantly expressed in leaf and reproductive buds during the growing season (Hsu et al., 2011). This suggests that PIF4, which functions as a direct upstream regulator of FT2 transcription, and PHYB, which modulates PIF4 protein stability through direct protein-protein interaction, may exhibit overlapping spatial expression patterns with FT2. Therefore, the authors should investigate the spatial expression patterns of PHYB and PIF4 in poplar by generating appropriate reporter lines or through *in situ* hybridization.

This is indeed an important point. In *Arabidopsis*, *FT*, *PIF4* and *PHYB* all have overlapping expression patterns in minor leaf veins (ref 30,64). We expect this regulatory paradigm to be conserved in *Populus* leaves. However, characterizing promoters in trees presents significant technical challenges: (1) complementation assays to validate regulatory regions are exceptionally difficult, and (2) generating and analyzing promoter-reporter lines would require at least two years. Our own efforts to map regulatory regions around *FT2a* and *FT2b* loci have been ongoing for over four years. Additionally, *in situ* hybridization in poplar leaves is technically challenging and particularly challenging for *FT2* due to its low expression levels - a well-documented issue even in *Arabidopsis*.

Nevertheless, our whole-leaf expression data in aspen reveal highly coordinated patterns of *PHYB*, *PIF4* and *FT2* expression, strongly suggesting functional interaction in vegetative tissues. To further validate this, we conducted an RT-PCR analysis comparing major veins and leaf blades containing minor veins. The results clearly demonstrate overlapping expression of *FT2*, *PIF4* and *PHYB* in tissues with minor veins, mirroring the *Arabidopsis* pattern. This new data are presented in Figure S7B and discussed in lines 577-583.

Major #5. The authors mentioned ‘significantly’ on line 192, 195, 198, and 211 to describe the effect of PHYB or PIF4 on the regulation of growth cessation. However, the statistical significance of the data has not been confirmed. The authors should ensure that all claims of statistical significance are supported by appropriate analyses and validation (Fig. 1A, B, E, and F, Fig. S1B-G, Fig. 2A, B, D, and E, Fig. S2A and B, Fig. 3B, Fig. S3C, Fig. 4D and E, Fig. S4C and D, Fig. 5A and B, Fig. S6A). Additionally, in many

cases, the figure legends do not specify the statistical methods used to analyze the data (Fig. 4B, Fig. S4B, Fig. S5B, Fig. 6A, Fig. S6C). Providing this information is crucial for ensuring transparency and reproducibility. The authors should include details on the statistical tests performed.

We have now performed statistical analyses for all phenotypic and gene expression data and have included these results in each corresponding figure. The details of the statistical methods used are also included in the figure legends, which are all marked red.

Minor comments:

1. The figure legend for Fig. 4D should clearly specify the plant growth conditions.

We have now included the details of the growth conditions in the figure legend for Fig. 4D which is now 4E in the revised manuscript, line 341.

2. Line 347-349. The authors should support this claim with appropriate statistical analyses and rephrase it in a scientifically precise manner.

We performed statistical analysis on the Figure S3C dataset, which showed that WT plants grew significantly less at 15°C LD than at 21°C LD, whereas *PIF4a-YFP* overexpressing plants demonstrated significantly greater growth at 15°C compared to 21°C (Figure S3C). Please see the rephrased lines 372-375.

3. Line 519-520. Please provide the relevant data or include a reference to support this statement.

We have now generated additional experimental evidence supporting this conclusion. The newly added Figure S7A presents diurnal expression patterns of both FT2 genes under long-day conditions, clearly demonstrating their significant suppression at low temperatures. Furthermore, we have substantially expanded our discussion of the biological implications of this temperature-mediated FT2 downregulation, including the specific involvement of the PHYB-PIF4 module, lines 566-575.

Reviewer #2 (Remarks to the Author):

One major comment that I have is the E-box discovery. In general a discovery of this importance that can explain why PIF4 behaves different in poplar than arabidopsis could be more focused on. While indeed the yeast 1 hybrid show that PIF4 can bind the E-box, this is not definite and a ChIP and/or EMSA can be performed to show that PIF4 can indeed bind these regions in planta. This should be fairly straight forward especially as the authors already generated the PIF4-YFP OX lines.

Thank you for this excellent suggestion. Following your recommendation, we performed CUT&Tag analysis using leaves from PIF4a-YFP OE3 grafted plants. qPCR analysis confirmed significant enrichment at fragments containing the two E-box regions in *PIF4a-YFP OE3* plants relative to WT self-grafted controls (Figure 4C), while control loci showed no enrichment. These results prove that PIF4a binds specifically to the *proFT2_{E-box}* region *in planta*. We have incorporated these new findings in the revised manuscript (lines 303–310 and 771–794 in the Methods section).

Do the authors know if such Cis element also exist in Arabidopsis plants that are adapted to cold temperature? or is this a specific adaptation for poplar?

Here we would like to clarify our interpretation. Our data indicate that the E-box mediates temperature responsiveness via PIF4 binding, but these elements are not exclusive to PIF4, they likely function as general binding sites for PIF transcription factors.

Studies in Arabidopsis (Takagi et al., 2023) have characterized *FT*-regulatory regions without identifying G-box elements in the upstream promoter, consistent with our findings. The sole experimentally confirmed G-box, located 1.5 kb downstream of Arabidopsis *FT gene*, binds PIF4 (Zikola et al., 2019). Further, Kumar et al. (2016) demonstrated PIF4 binding near the *FT* start codon, where E-boxes are enriched (Zikola et al., 2019). Other thermoresponsive regulators (e.g., SVP, FLM, FLC) also target this region. Our findings proved that PIF4 binds to an intergenic region of two *FT2* paralogs in poplar trees.

That said, PIF4-mediated *FT* regulation at lower temperatures appears to be a *Populus*-specific adaptation, potentially linked to its role in growth cessation and bud set. We have expanded this discussion in the revised text to provide further context, lines 585–594.

Also as PIF4 is classified as a warm temperature signaling protein in Arabidopsis, does it play any role in warm temperature in poplar?

This is an intriguing question. The effects of warm temperatures on growth cessation and bud formation in poplar trees have been reported with some inconsistency. For instance, Lee et al. (2009) found that warm temperatures accelerate growth cessation under short photoperiods, whereas Rohde et al. (2011) observed a delay in growth cessation at higher temperatures. In our experiments with PIF4 mutant trees compared to WT, under short-day (SD) conditions and elevated temperatures (28°C), we noticed slightly earlier growth cessation but delayed bud formation (data not shown), suggesting that PIF4 plays a role in the warm temperature response.

These findings highlight the complexity of how poplar trees respond to warm temperatures, underscoring the need for a more detailed characterization of PIF4's function. In this study, we fully investigated PIF4's role in colder climates (21°C vs.

15°C). While PIF4 likely operates in a similar manner under warm temperatures, such as through increased transcriptional and protein levels, its precise biological effects in warm conditions remains to be explored.

A minor comment is in the way figures are mentioned in text, its better to sperate the letters such as Fig1A,B (See line 133 for example)

We now have changed all the figure letters according to the reviewer's suggestion.

REVIEWERS' COMMENTS

Reviewer #1 (Remarks to the Author):

The authors have addressed most of my concerns, which I appreciate. However, one important issue remains unresolved and should be corrected to strengthen the conclusion of this manuscript.

While the authors present Western blot results in Figure S1J to support protein-level changes, the quality of the blot image is questionable. The bands appear inconsistent, especially for the loading control (RPNP6), making it difficult to assess the reliability of normalization and quantification. I strongly recommend providing a higher-quality blot with clearly consistent loading control signals to strengthen the reliability of the data.

We appreciate the reviewer's observations regarding the Western blot results in Figure S1j. We acknowledge that the data quality was suboptimal for some samples, likely due to technical challenges associated with analyzing soil-grown SD leaf samples. The observed variability in loading controls, particularly in ZT16 samples, likely reflects interference from secondary metabolite accumulation in SD leaf tissues, potentially compromising both protein extraction efficiency and detection sensitivity. Given these considerations, we have reanalyzed the blots, obtaining similar results that are now incorporated into the revised version.

Since we do not have any more samples from this experiment, and after thorough consideration, we suggest to exclude the ZT16 time point data from our analysis, as the loading controls were especially variable for these samples and their removal does not affect the overall conclusions of our study. Importantly, despite these technical limitations, comparative analysis with wild-type controls allowed for reliable detection of the target bands.

To provide additional experimental validation, we performed an independent experiment using in vitro-grown samples under SD conditions (Figure S1k). These controlled experiments robustly confirmed the temperature-dependent reduction in PHYB2-GFP abundance, further validating the patterns observed in Figure S1j. This complementary approach strengthens the reliability of our findings while addressing the technical limitations noted in the original soil-grown samples.

Additional comments:

The authors have thoroughly addressed the concerns raised by Reviewer 2. They have offered a clear and well-reasoned explanation, strengthening the manuscript's clarity. I find their responses are convincing and have no further comments.

We appreciate the reviewer's positive comments regarding the revised manuscript.